# Multiple Autonomous Cell Death Suppression Strategies Ensure Cytomegalovirus Fitness

**DOI:** 10.3390/v13091707

**Published:** 2021-08-27

**Authors:** Pratyusha Mandal, Lynsey N. Nagrani, Liliana Hernandez, Anita Louise McCormick, Christopher P. Dillon, Heather S. Koehler, Linda Roback, Emad S. Alnemri, Douglas R. Green, Edward S. Mocarski

**Affiliations:** 1Emory Vaccine Center, Department of Microbiology and Immunology, Emory University School of Medicine, Atlanta, GA 30322, USA; liliana.hernandez@emory.edu (L.H.); heather.s.koehler@emory.edu (H.S.K.); lroback@emory.edu (L.R.); 2Pharmaceutical Product Development, Wilmington, NC 28401, USA; lynsey.nagrani@gmail.com; 3BioMarin Pharmaceuticals, Novato, CA 94949, USA; alouisemccormick@gmail.com; 4Pfizer, San Diego, CA 92121, USA; dillon.christopher@gmail.com; 5Department of Biochemistry and Molecular Biology, Thomas Jefferson University, Philadelphia, PA 19107, USA; Emad.Alnemri@jefferson.edu; 6Department of Immunology, St. Jude Children’s Research Hospital, Memphis, TN 38105, USA; douglas.green@stjude.org

**Keywords:** BCL2 family, mitochondria, serine protease, herpesvirus, apoptosis, extrinsic apoptosis, intrinsic apoptosis, pyroptosis, necroptosis, inflammation, TNF, IFN, DNA virus, cytomegalovirus, replication, HCMV, MCMV, cmvPCD, UL36, M36, vICA, M45, vIRA, M38.5, M41.1, vMIA, vIBO Caspase-8, RIPK1, RIPK3, myeloid cells, macrophages, cell death

## Abstract

Programmed cell death pathways eliminate infected cells and regulate infection-associated inflammation during pathogen invasion. Cytomegaloviruses encode several distinct suppressors that block intrinsic apoptosis, extrinsic apoptosis, and necroptosis, pathways that impact pathogenesis of this ubiquitous herpesvirus. Here, we expanded the understanding of three cell autonomous suppression mechanisms on which murine cytomegalovirus relies: (i) M38.5-encoded viral mitochondrial inhibitor of apoptosis (vMIA), a BAX suppressor that functions in concert with M41.1-encoded viral inhibitor of BAK oligomerization (vIBO), (ii) M36-encoded viral inhibitor of caspase-8 activation (vICA), and (iii) M45-encoded viral inhibitor of RIP/RHIM activation (vIRA). Following infection of bone marrow-derived macrophages, the virus initially deflected receptor-interacting protein kinase (RIPK)3-dependent necroptosis, the most potent of the three cell death pathways. This process remained independent of caspase-8, although suppression of this apoptotic protease enhances necroptosis in most cell types. Second, the virus deflected TNF-mediated extrinsic apoptosis, a pathway dependent on autocrine TNF production by macrophages that proceeds independently of mitochondrial death machinery or RIPK3. Third, cytomegalovirus deflected BCL-2 family protein-dependent mitochondrial cell death through combined TNF-dependent and -independent signaling even in the absence of RIPK1, RIPK3, and caspase-8. Furthermore, each of these cell death pathways dictated a distinct pattern of cytokine and chemokine activation. Therefore, cytomegalovirus employs sequential, non-redundant suppression strategies to specifically modulate the timing and execution of necroptosis, extrinsic apoptosis, and intrinsic apoptosis within infected cells to orchestrate virus control and infection-dependent inflammation. Virus-encoded death suppressors together hold control over an intricate network that upends host defense and supports pathogenesis in the intact mammalian host.

## 1. Introduction

In the arms race between eukaryotic hosts and pathogens, programmed cell death (PCD) mechanisms evolved as a network of pathways primarily intended to sustain health by resisting infection [1,2,3,4,5]. PCD pathways play roles in host defense against intracellular pathogens by eliminating cells before the production of progeny occurs. A primary form of PCD is apoptosis, a caspase (CASP)-dependent death pathway initiated either by infection-associated external danger signals (extrinsic) or by intracellular stress sensed via mitochondria (intrinsic) [5,6,7]. An alternate form of cell death, necroptosis, unleashes in the face of virus-encoded CASP inhibitors and has been characterized as a highly potent host defense pathway known to limit the ability of a pathogen to invade the host [8,9,10]. Cytomegalovirus (CMV) has evolved to be a ubiquitous, highly successful mammalian pathogen by employing an array of conserved immunomodulatory strategies that subvert anti-viral defenses, particularly through the effective suppression of both extrinsic and intrinsic PCD pathways [10,11,12,13,14,15,16,17,18,19]. These suppression strategies sustain human (H) and murine (M) CMV replication in cells and, most importantly, ensure fitness in vivo [10,12,13,14,15,16,17,18].

HCMV is a significant pathogen with protean clinical manifestations where pathogenesis depends on viral replication as well as immune cell-driven inflammation [20,21,22,23,24,25,26,27]. The contribution of PCD suppression to pathogenesis remains poorly understood. Studies in MCMV have revealed similarities to HCMV in pathogenesis, host-pathogen interactions, and antiviral host defense mechanisms [10,17,23]. We, and others, have identified distinct, conserved pathways that mold the inflammatory environment through crosstalk in addition to restricting MCMV replication [28,29,30,31,32]. All known DNA viruses encode inhibitors of CASP8 [5]. The viral inhibitor of caspase activation (vICA) suppresses innate inflammatory signaling in human or murine cells to sustain MCMV fitness [30,33,34]. Both HCMV and MCMV also suppress mitochondrial cell death by targeting the B cell lymphoma-2 (BCL2)-associated X protein (BAX) and BCL-2 homologous antagonistic killer (BAK) during infection [17,31,32,35,36,37,38]. Here, we utilized MCMV infection to investigate how combined BAX and BAK suppression by the MCMV-encoded suppressors impact pathogenesis as well as how mitochondrial signaling interfaces with other PCD pathways.

As cellular stress is conveyed to the mitochondria, monomeric BAX and BAK homo-oligomerize at the mitochondrial outer membrane (MOM), driving a well-defined PCD outcome [39,40,41] where BAX/BAK oligomers drive increased mitochondrial membrane permeability resulting in the release of cytochrome c, reactive oxygen species, and HtrA2/Omi with a consequent activation of CASP9, Apaf-1, and executioner CASP3 and CASP7 [41,42,43]. BCL-2 family proteins BID, BIM, and PUMA may act upstream of BAX and BAK [44]. All large DNA viruses encode suppressors of mitochondrial cell death [45,46]. Mitochondria also control CASP-independent death pathways [47,48] such as the HtrA2/Omi-dependent pathways that are suppressed by the viral inhibitor of mitochondrial apoptosis (vMIA) during HCMV replication [13,49]. Related suppressors are encoded by adenovirus E1B [50,51], vaccinia virus F1L [52], myxoma virus M11L [53,54], and gammaherpesvirus vBCL-2 homologs [55]. Activated BID bridges extrinsic death signaling with BAX/BAK-dependent mitochondrial cell death, an amplification step that may be crucial in cells where CASP8 levels are too low to directly cleave CASP3 and/or CASP7 [56,57,58]. This amplification contributes markedly to extrinsic death signaling through the TNF-TNFR1-CASP8 signaling axis. During HCMV infection, gene product UL37x1-encoded vMIA suppresses the activation of BAX as well as BAK [16,59]. Thus, vMIA suppresses caspase-dependent as well as caspase-independent forms of PCD, ensuring cells survive and produce viral progeny. MCMV encodes two related cell death suppressors: M38.5-encoded vMIA that targets the BAX [17,36,60,61] and M41.1-encoded viral inhibitor of BAK oligomerization (vIBO) [32,38,62,63]. Individually, vMIA- or vIBO-deficient MCMV exhibit a modest impact on pathogenesis [31,32,38,61,62,63], consistent with the target protein specificity and the known overlap in BAX and BAK function in contrast to the broader function of HCMV vMIA. Studies with single mutants disrupting vMIA or vIBO function reveal that BAX or BAK suppression occurs in a cell type-dependent fashion [30,31,32]. The combined effect of vMIA and vIBO on virus fitness, replication, and pathogenesis, as well as crosstalk with innate cytokine activation remains to be investigated.

Here we showed that combined BAX/BAK suppression by vMIA/vIBO prevents intrinsic PCD, limits HtrA2/Omi impact, and regulates innate inflammation. vMIA/vIBO remain essential for sustained MCMV titers in the salivary glands of mice, the tissue known to mediate horizontal transmission of CMV. Macrophages represent a relevant, natural setting where MCMV-encoded PCD suppressors show functions that correspond to behavior in infected hosts [30]. Macrophages and their monocyte precursors play central roles during CMV pathogenesis by sustaining viral replication to support reservoirs that underly life-long persistence and latency [64,65,66,67,68,69,70,71]. We showed that MCMV faces receptor interacting protein (RIP) kinase (RIPK)3-dependent necroptosis within ~6 to 9 h, TNFR1-dependent CASP8 activation within ~12 h, and mitochondrial cell death within ~24 h of infection. Thus, PCD pathways are temporally staged and the suppression of each PCD pathway proceeds independently based on studies that rely on mutant macrophages that lack specific cell death components. MCMV gene products, M45-encoded viral inhibitor of RHIM activation (vIRA), M36-encoded vICA, and M38.5/M41.1 encoded vMIA and vIBO work in a concerted manner during infection with each suppressor playing a largely non-redundant role in sustaining viability, facilitating virus replication, and orchestrating infection-associated inflammation. vMIA/vIBO also suppresses the amplification of TNF signaling via mitochondria and likely underlies observations showing M36 mutant MCMV does not activate this sub-pathway [72]. The interfacing of the mitochondrial suppressors with the TNF signaling brings to light new insights regarding the role of this inflammatory cytokine in CMV biology, revealing impacts on replication in a variety of cell types as well as pathogenesis in vivo [73,74,75,76].

Given that vMIA/vIBO function plays out independently of extrinsic apoptotic or necroptotic players, MCMV employs sequential, specific suppression mechanisms with non-overlapping functions to establish a protective shield against host immunity. This prevents viral clearance and ensures fitness. Our work sheds light on the potential for targeting the CMV-encoded PCD suppressors as well as the associated host pathways as intervention strategies to reduce pathogen load and disease pathogenesis.

## 2. Materials and Methods

### 2.1. Cell Culture and Reagents

3T3-SA fibroblasts (ATCC CCL-92), SVEC4-10 endothelial cells (ATCC CRL-2181), and 293-T cells were maintained at 37 °C with 5% CO_2_ in Dulbecco’s modified Eagle medium (DMEM) containing 4.5 g/mL glucose, 1 mM sodium pyruvate, 10% fetal bovine serum (Atlanta Biologicals, Oakwood, GA, USA), 2 mM L-glutamine, 100 U/mL penicillin, and 100 U/mL streptomycin (Invitrogen, Waltham, MA, USA). BMDM were generated as described previously [72]. Briefly, flushed marrows from tibias and femurs of 8- to 12-week-old mice were cultured for 7 days in DMEM containing 4.5 g/mL glucose (10-013 CV, Corning, Charlotte, NC, USA), 10% fetal bovine serum (F2442, Sigma-Aldrich, St. Louis, MO, USA), and 2 mM l-glutamine (MT 25005CI, ThermoFisher Scientific, Marietta, GA, USA) supplemented with 100 units/mL penicillin and 100 units/mL streptomycin (MT 3002CI, Thermo Fisher Scientific, Waltham, MA, USA). For BMDM culture, this complete DMEM medium had a final 20% fetal bovine serum and 20% filtered L929-conditioned medium (as a source of macrophage colony-stimulating factor). Cells were harvested from C57BL/6 (referred to as WT), *Ripk3^−/−^ Casp8^−/−^, RIPK3^−/−^*, *Ripk1^−/−^Casp8^−/−^Ripk3^−/−^*, *Ripk1^K45A/K45A^*, *Ripk3^K50A/K50A^*, *Asc^−/−^*, *Aim2^−/−^*, *Ifnar1^−/−^*, *Ifngr^−/−^,* and *Gsdmd^−/−^* [77], *Zbp^−/−^* [78], *Bak^−/−^Bax^fl/fl^* (also referred to as *Bax^tm2Sjk^ Bak1^tm1Thsn/J^*) [38], *Bid^−/−^*, *Bim^−/−^*, *Puma^−/−^*, *Bid^−/−^Bim^−/−^Puma^−/−^* [79], *Bax^−/−^Bak^−/−^* [80], and *Bok^−/−^* [81] mice. BAX knockdown in *Bak^−/−^Bax^fl/fl^* BMDM cells was achieved by infection with lentivirus expressing Cre recombinase. Lentiviral stocks were generated by transfecting 293-T cells with LV-Cre or control LV-GFP (Addgene, Watertown, MA, USA) along with packaging plasmids psPAX2 and VSV-G [82]. Supernatant containing the lentiviral particles was collected 24, 48, and 72 h post transfection, filtered through a 0.45-micron filter, subjected to a flash-freeze, and stored at −80 °C until use. WT and *Bak^−/−^Bax^fl/fl^* BMDM cells were thawed and cultured in the lentiviral supernatant collected at 24 h post transfection for 8 h. Medium was replaced and the BMDM cells were allowed to recover overnight. This process was repeated twice using supernatant collected at 48 and 72 h post transfection. Following infection with lentiviral particles, BMDM cells were allowed to rest for two days before seeding. For TNF treatment of cells, 25 ng/mL of recombinant murine TNF (315-01A, Peprotech Inc, Cranbury, NJ, USA) was added 1 hpi with viruses. All cells were maintained at 37 °C in a humidified 5% CO_2_ incubator.

### 2.2. BAC Mutagenesis and Recombinant Viruses

*Escherichia coli* strain DH10B harboring plasmid pSIM6 encoding the λ red recombineering functions [83] and MCMV strain K181-BAC [84] were utilized to introduce insertion mutations in the viral genome. To generate a virus lacking both of these mitochondrial cell death suppressors, a vIBO-deficient virus that was generated and characterized previously, m41.1.ΔStart-BAC [38], was employed. A selection/counterselection cassette, Kan^R^/SacB, was inserted into M41.1.ΔStart-BAC within the m38.5 reading frame and upstream of overlapping M38. This virus was named ΔM38.5/M41.1-BAC, signifying the presence of large insertion disrupting m38.5 and a point mutation in M41.1. The following primers were used to insert the Kan/SacB cassette: M38.5 Kan/SacB-F(5′ccaagcgccccagaggcgaagagcagcgctggtcgttcgcttacaaacccaattcgagctcggtacccgg); M38.5 Kan/SacB-R(5′ggttgtagttgtggaggggacagcgatggagagtgtgcgccgacccttctatcccgggaaaagtgccacc).

A second double mutant virus was generated to confirm the results produced by ΔM38.5/m41.1-BAC and to identify differences that may be due to the insertion cassette disrupting M38.5 as well as neighboring genes. The second double mutant virus, M38.5.StopFS/M41.1.ΔStart-BAC, was also generated from M41.1.ΔStart-BAC with the more efficient En Passant mutagenesis system [85,86]. A point mutation was engineered into M38.5, creating a stop codon and frame shift mutation just eight amino acids from the start of the protein. Furthermore, the point mutation created an XbaI restriction site used for screening and characterization of the mutant virus. The following primers were used to insert this point mutation: M38.5.StopFS-F(5′gcgccccagaggcgaagagcagcgctggtcgttcgcttacaaaccctctagaagggtcggcgcacaggatgacgacgataagtaggg) and M38.5.StopFS-R(5′gtagttgtggaggggacagcgatggagagtgtgcgccgacccttctagagggtttgtaagcgaaccaaccaattaaccaattctgattag). Finally, a single M38.5 mutant virus, M38.5.StopFS-BAC, was generated from the parental K181-BAC with the same point mutations using the primers described above.

All BAC-derived clones were screened by sequencing PCR products over the M41/M41.1 locus as previously described [38] as well as over the M38.5 locus. The following primers were used to amplify and screen the M38.5 locus: M38.5-F(5′gcagaagtcacgtcggatccag); m38.5-R(5′ggctgctacgagaacgtgac). Furthermore, BAC-derived full-length clones were analyzed by RFLP with five restriction enzymes to ensure the integrity of the viral genome. Following genetic characterization, mutant and parental viruses were reconstituted and plaque-purified as previously described [38]. Growth curves and viral yields were performed by infecting cells in a 48-well plate in 0.25 mL of viral inoculum at the indicated multiplicity of infection (MOI) for 1 h at 37 °C with 5% CO_2_. Following adsorption, the inoculum was replaced with complete DMEM. Samples were harvested at the indicated times, sonicated, and titered by plaque assay on 3T3-SA fibroblasts.

### 2.3. Mice, Infections, and Organ Harvests

BALB/c, B6 × 129 P2, and C57BL6/J mice were obtained from Jackson Laboratory (Bar Harbor, ME USA). B6 × 129-*Bax^tm2Sjk^ Bak1^tm1Thsn/J^* (also referred to as *Bak^−/−^Bax^fl/fl^*) were obtained and used as previously described [38]. *Casp8^−/−^RIPK3^−/−^* were bred and maintained as previously described [87]. Six-week-old BALB/c mice were infected via the intraperitoneal (IP) route of inoculation with 2 × 10^5^ PFU of tissue culture-derived virus. Other strains of mice (6–18 weeks old) were infected via the IP route or via footpad route with 1 × 10^6^ PFU or 5 × 10^6^ PFU of virus respectively. At the time of sacrifice, organs were placed in 1 mL of complete DMEM, disrupted by sonication, and virus titer was determined by plaque assay as described [30]. IE1 positive cells were identified as described [30].

### 2.4. Cell Viability Assay

At the indicated times, cell viability was determined by measuring the ATP levels using the Cell Titer-Glo Luminescent Cell Viability Assay kit (Promega Corporation, Madison, WI, USA) according to manufacturer’s protocol. Cells (~50,000 cells/well) were seeded into 96-well plates in triplicate. Approximately 18 h post seeding, medium was replaced with 100 μL of virus inoculum containing 10 PFU/cell (MOI = 10). At 24 hpi, and DNA fragmentation was visualized by the DeadEnd Fluorometric TUNEL System (Promega Corporation, Madison, WI, USA). Cells (60,000/well) were seeded into 48-well plates in triplicate. Approximately 18 h post seeding, cells were infected at MOI = 10. The number of cells with detectable fragmented DNA were expressed as a percent of cells expressing IE1. To assess cell morphology, cells were plated in 48-well plates as indicated above and infected with virus at MOI = 20 at 18 h post plating. Phase-contrast microscopy was performed 24 hpi. To assess the inclusion of cell permeable SYTOX, Green dye (S7020, Invitrogen, Waltham, MA, USA) uptake was used in real time using an IncuCyte instrument (Essen BioScience, Ann Arbor, MI, USA) as described [82]. BMDM were plated at 5 × 10^4^ cells per well in 24-well tissue culture plates. Cells were infected as described above at MOI = 10.

### 2.5. Immunoblot Analysis

To detect BAX quantities, cells were lysed in 50 mM Tris (pH 7.0), 2% sodium dodecyl sulfate (SDS), 5% β-mercaptoethanol, and 2.75% sucrose. Cell lysates were boiled for 5 min, separated by SDS-polyacrylamide gel electrophoresis (15% gel), transferred to Immobilon-P PVDF membranes (Millipore Sigma, Burlington, MA, USA), and subjected to immunoblot analysis. For identification of CASP activation patterns, infected cells were harvested at 24 hpi and evaluated as described previously [29]. Briefly, 5 × 10^6^ BMDM were plated in a 10-cm tissue culture dish. Twenty-four hours post-plating, cells were infected with the virus. Lysates were harvested utilizing 1000 μL of ice-cold Triton-X lysis buffer supplemented with protease and phosphatase inhibitors for 30 min. Ultracentrifuged (150,000 rpm at 4 °C for 20 min) lysates were separated from Triton-X insoluble pellets. Pellets were sonicated for 10 s at 20 watts in 150 μL of disruption buffer (50 mM Tris pH 6.8, 5% β-mercaptoethanol, 2.75% [*w*/*v*] sucrose and 2% [*w*/*v*] SDS). Given the comparable sizes of the caspases (for both total and cleaved forms), individual immunoblotting was performed for each caspase. Loading control actin was detected for each caspase blot individually. Antibodies for immunoblots: anti-BAX (BioLegend, SanDiego, CA, USA; clone 5B7 (currently discontinued by manufacturer)); rabbit anti-CASP9 (9504), rabbit anti-cleaved CASP8 (8592), rabbit anti-cleaved CASP7 (8438), rabbit anti-cleaved CASP3 (9661), (all from Cell Signaling Technology, Denvers, MA, USA), and mouse anti-β-actin (A2228, Sigma-Aldrich, St. Louis, MO, USA).

### 2.6. Cytokine Array

Released cytokines and chemokines were detected by cytokine array (Cytokine Array Panel A, ARY006, R&D Systems, Minneapolis, MN, USA). Supernatants were collected from cultures of WT BMDM infected with MCMV (K181 or mutant) for 24 h. Following centrifugation at 500 g for 5 min at 4 °C, cell-free supernatants were pooled from two independent experiments for each condition. Array strips were exposed to chemiluminescent chemicals (provided with the kit) and raw images were developed on X-ray film. Films were analyzed using Adobe Photoshop 8 to generate pixel intensity of dots corresponding to individual cytokines as well as reference dots. Pixel intensities of control dots were averaged for each array strip. For comparing four array strips (heat map in Figure 3), reference spots were normalized across the dot blots.

### 2.7. Statistics

All statistical analyses were performed using GraphPad Prism 8 Software (GraphPad Software Inc. La Jolla, CA, USA). Two groups of data were compared for significance using non-parametric, Wilcoxon matched-pairs signed rank test; three or more groups of data were compared using ANOVA. All graphs show standard error and mean for datasets, except viral titers in organs where mean is indicated for each data set. *p*-values (*p*) of <0.05 were considered significant and indicated with * <0.05, ** <0.01, *** < 0.001, **** <0.0001.

### 2.8. Software

Chemiluminescent images of immunoblots were captured using the Kwik Quant system (Kindle Biosciences, Greenwich, CT, USA) and analyzed on Adobe Photoshop 8 (Adobe Corporation Inc., San Jose, CA, USA). All cell death as well as ELISA data were assembled using Microsoft Excel (Microsoft Corporation, Redmond, WA, USA) and graphed using GraphPad Prism 8 (GraphPad Software Inc. San Diego, CA, USA). For cytokine array, pixel intensities were determined using Adobe Photoshop 8. All figures were assembled using Adobe Illustrator 8. Software Inc. San Diego, CA, USA). Model cartoons were constructed using BioRender software (BioRender, Toronto, ON, Canada).

### 2.9. Data Availability

Further information and requests for resources and reagents should be directed to and will be fulfilled by mocarski@emory.edu and pratyusha.mandal@emory.edu. All data are included in manuscript. All raw files are available upon request. This manuscript generated a new MCMV mutant (∆M38.5/M41.1 double mutant virus) that is available upon request. All reagents utilized are described in detail in these methods.

### 2.10. Study Approval

All experiments were conducted with approval (4 February 2019, Proto 201700351) from the Emory University Biohazard, Chemical Hazard Review and Animal Use (IACUC) Committees.

## 3. Results

### 3.1. M38.5-Encoded vMIA and M41.1-Encoded vIBO Support Viral Fitness

To determine the impact of combined BAX/BAK suppression by M38.5-encoded vMIA and M41.1-encoded vIBO, we compared the replication of mutant viruses to parental K181 (Figure 1A–C). The genomic integrity of double mutants and control viruses were confirmed by direct evaluation (Appendix A), as previously shown for single mutant viruses [29,30,38]. MCMV was shed in saliva mediates horizontal transmission [88], so viral load in these glands may therefore be considered an indicator of viral fitness. MCMV detection in salivary glands requires entry followed by replication in cells at the origin of infection, myeloid-cell mediated dissemination to different tissues including the salivary glands, and ultimate replication salivary glands [88,89,90,91,92,93]. Virus-encoded modulators of adaptive and innate immunity (including cell death suppressors) suppress host defense to influence infection at different stages in dissemination [30,93,94]. We determined the titers of mutant viruses described in the salivary glands of BALB/c mice inoculated intraperitoneally and C57BL6/J mice inoculated via footpad at 14 days post infection (dpi) (Figure 1B,C). At 14 dpi, vMIA/vIBO-deficient viruses (ΔM38.5/M41.1 and M38.5.StopFS/M41.1.ΔStart) showed titers that averaged ~1000-fold lower than K181 in the salivary glands of highly susceptible BALB/c mice (Figure 1B). Viral titers were markedly more variable and lower in most animals than observed with single mutant viruses (M38.5.StopFS or M41.1.∆Start), which exhibited a modest compromise compared with K181 [30,38]. The variability of double mutant viruses contrasts the invariant titers of K181, or single mutants, reflecting the importance of BAX and BAK suppression to sustaining MCMV fitness. All mutants (single or double) remained severely compromised when titers were assessed in salivary glands of C57BL6/J mice following footpad inoculation (Figure 1C). Together, these observations revealed that the vMIA/vIBO double mutant MCMV is attenuated for dissemination to or replication in salivary glands, highlighting the importance of BAX/BAK suppression for viral fitness in vivo.

To better understand the infection pattern in different cell types when BAX/BAK were not suppressed, we evaluated the replication of double-deficient viruses in cultured primary macrophages, and immortalized fibroblasts as well as endothelial cells, all of which support parental K181 replication but show differential susceptibility to various MCMV mutants [30] consistent with cell type-specific function of virus-encoded cell death suppressors (Figure 1D–F and Appendix A). ∆M38.5/M41.1 and M38.5.StopFS/M41.1.ΔStart viruses showed severe attenuation in BMDM from wild type (WT) C57BL/6J mice (Figure 1D), a pattern that was confirmed for ∆M38.5/M41.1 over a nine-day time course (Appendix A). Double-mutant virus did not increase in titer over this time course. Surprisingly, the infection of ∆M38.5/M41.1 in BMDM exhibited reduced IE1+ cells over MOI range from 0.626 through 10 (Appendix A) at 24 h post infection (hpi), even though no defect in IE1 expression was observed for either M38.5 or M41.1 single-mutant viruses [31,32]. Importantly, the M38.5/M41.1 double-mutant viruses were constructed utilizing the original K181 background strain (Figure 1A) that displays no defect in entry for myeloid cells, unlike other pSM3fr-derived MCMV [95]. The reason vMIA/vIBO mutant MCMV shows this defect will require further investigation. Immortal 3T3-SA fibroblasts exhibited modestly lower vMIA/vIBO mutant viral titers at 24 through 72 hpi even though titers increased by 5 dpi to match K181 (Figure 1E and Appendix A). SVEC4-10 cells exhibited comparable levels of K181 and mutant virus replication (Figure 1F and Appendix A). Thus, in culture settings, mitochondrial suppression by vMIA and vIBO contributes most dramatically to viral replication in macrophages.

### 3.2. HtrA2/Omi Limits MCMV Replication in Macrophages

The HCMV vMIA-deficient virus has been studied in human fibroblasts, where vMIA suppresses BAX/BAK-dependent mitochondrial apoptosis, as well as mitochondrial serine protease HtrA2/Omi-dependent death [13]. HtrA2/Omi terminates HCMV infection by inducing CMV-dependent PCD (cmvPCD), which is unleashed earlier by the vMIA-deficient virus [13]. To determine how mitochondrial HtrA2/Omi interfaces with MCMV vMIA and vIBO or its function during MCMV infection of macrophages, we infected WT or *Htra2/Omi^−/−^* BMDM with K181 or ∆M38.5/M41.1 at two different MOIs (0.5 or 5; Figure 1G). K181 titers were ~50- to 100-fold higher in *Htra2/Omi^−/−^* BMDM compared with WT cells, indicating that this serine protease restricts MCMV infection as it does HCMV infection [13]. The elaboration of necroptotic, extrinsic apoptotic and mitochondrial death suppressors (vIRA, vICA, and vMIA/vIBO respectively) failed to completely prevent HtrA2/Omi-induced restriction of MCMV replication. The vMIA/vIBO double deficient virus exhibited 5 to 10-fold higher titer in *Htra2/Omi^−/−^* BMDM, even though the mutant virus remained severely attenuated in both WT and *Htra2/Omi^−/−^* cells. Thus, HtrA2/Omi contributes to but is not the primary host effector driving attenuation of the vMIA/vIBO double-deficient virus. During MCMV infection of macrophages, the combined action of vMIA/vIBO suppress mitochondrial signaling, including HtrA2/Omi, to benefit sustained virus replication.

### 3.3. vMIA and vIBO Suppress BAX/BAK-Dependent Cell Death

In BMDM, the replication defect of ∆M38.5/M41.1 mutant virus became evident by 72 hpi (Appendix A). In line with these observations, the vMIA/vIBO double-deficient MCMV sensitized macrophages to loss of viability when compared with K181 infection by 48 hpi (Figure 2A). In ∆M38.5/M41.1 infection settings, a significantly elevated proportion of IE1(+) cells exhibited DNA fragmentation by TUNEL assay when compared with K181 infection at 24 and 48 hpi (Figure 2B). Both infection settings retained a comparable proportion of IE1(+) cells without DNA fragmentation. At 24 hpi with ∆M38.5/M41.1, ~3–5% of IE1(−) cells exhibited DNA fragmentation in comparison with ~0.5–1% of K181 infected cells. Thus, the combined effects of vMIA/vIBO contribute to maintaining nuclear integrity during infection. Viability appeared to be rescued, albeit modestly in *Bax^−/−^Bak^−/−^* but not *Bax^fl/fl^Bak^−/−^* BMDM (Figure 2C), suggesting that combined BAX and BAK suppresses MCMV-induced death of macrophages. Cell morphology assessed at 24 hpi by phase contrast microscopy revealed apoptotic morphology with cellular blebbing during ∆M38.5/M41.1 infection (Figure 2D). Therefore, vMIA/vIBO suppression protects macrophages from BAX/BAK-mediated apoptotic death.

Cre-recombinase-mediated knockdown of *Bax*, reduced BAX protein expression in *Bax^fl/fl^Bak^−/−^,* BMDM, and rescued replication of both double-mutant viruses, confirming that vMIA/vIBO-mediated suppression of BAX/BAK prevents cell death and facilities replication (Figure 2E and Appendix A). Rescue of replication was partial potentially due to incomplete elimination of BAX. Thus, both the viability and replication of ∆M38.5/M41.1 viruses were rescued by combined downregulation of BAX and BAK. Noteworthy, levels of cell death in WT, *Bax^fl/fl^Bak^−/−^*, *Bax^−/−^Bak^−/−^,* or *HtrA2/Omi^−/−^* cells infected with K181 all appeared similar. However, *Bax^fl/fl^Bak^−/−^* BMDM treated with LV-cre exhibited modestly increased K181 replication levels (Figure 2E). This pattern of rescue of viability in *Bax^−/−^Bak^−/−^* cells raises the possibility that factors not evident in these assays may contribute to the death initiated by vMIA/vIBO double-deficient MCMV. Given the recognized crucial functions of BID, BIM, PUMA, and BOK in activating BAX/BAK-dependent cell death, we assessed viability and viral titer in WT compared to *Bid^−/−^Bim^−/−^Puma^−/−^* as well as *Bid^−/−^*, *Bim^−/−^*, *Puma^−/−^,* and *Bok^−/−^* BMDM during infection with K!81 or ∆M38.5/M41.1 viruses (Figure 2F left panel and Appendix A). These single-mutant cells all exhibited a similar pattern of infection with either virus comparable to WT settings, demonstrating that, individually, BID, BIM, PUMA, or BOK are dispensable for suppression of ∆vMIA/vIBO-driven, BAX/BAK-mediated PCD (Appendix A–E). In contrast, *Bid^−/−^Bim^−/−^Puma^−/−^* cells exhibited a significant rescue in replication of either double mutant virus whereas *Bim^−/−^Puma^−/−^* cells showed a modest rescue (Figure 2F left panel). Together with Figure 1, these observations demonstrate that macrophages rely on mitochondria in responding to MCMV infection. The following steps occurred: HtrA2/Omi, together with BAX and BAK, restricts K181 infection even in the presence of vMIA and vIBO without influencing cell viability. In infected cells (Figure 2F right panel), combined functions of vMIA and vIBO primarily suppress BID-BIM-PUMA-BAX-BAK-dependent apoptosis and promote sustained replication.

### 3.4. vMIA and vIBO Suppress Contributions from TNF Signaling

During MCMV infection of macrophages, vMIA/vIBO suppress BAX/BAK-dependent mitochondrial death signaling, vICA suppresses TNF-CASP8-dependent apoptosis, and vIRA suppresses RIPK3-dependent necroptosis such that ∆M38.5/M41.1, ∆M36, and M45*mut*RHIM mutant viruses activate BAX/BAK-, TNF-CASP8-, and RIPK3-mediated pathways, respectively (Figure 3A). Whether these distinct patterns interface with each other during infection remains unclear. TNF signaling engages adaptor FADD-dependent CASP8 activation via TNFR1 independently of BID or BAX and BAK [29,73]. vICA suppresses CASP8 activation to avoid TNFR1-FADD-CASP8-dependent apoptosis and sustain fitness in vivo [29,30,73]. This TNF-dependent signaling is known to be active in control over HCMV replication as well as MCMV pathogenesis in vivo [74,75,76]. Currently, there is no understanding of how TNF contributes in the vMIA/vIBO suppression axis. WT and *Tnfr1^−/−^* BMDM were infected with K181, ∆M38.5/M41.1, necroptotic M45*mut*RHIM, or apoptotic ∆M36 virus for 24 or 48 h (Figure 3B). Remarkably, knockout cells completely resisted ∆M38.5/M41.1-induced cell death at 24 hpi, but became as susceptible as WT BMDM by 48 hpi, indicating that vMIA and vIBO suppressed an early contribution from TNF signaling to cell death. The addition of extraneous TNF to BMDM did not alter susceptibility of cells to mutant virus infection (Figure 3C). These observations affirm our previous findings that tonic TNFR1-dependent signaling is sufficient to dictate outcome during MCMV infection of macrophages [29]. The threshold was not amplified by the addition of extraneous cytokine. Necroptotic RIPK3 [30], pyroptotic mediators (CASP1 and CASP11) [96,97], pyroptosis executioner gasdermin D [98], MCMV-associated inflammasome components (adaptor ASC, nucleic acid sensor AIM2) [99], as well type I and type II interferon signaling (via interferon alpha/beta receptor 1 [IFNAR1] or interferon gamma receptor [IFNGR] respectively) were all dispensable for the BAX/BAK-dependent death induced by the vMIA/vIBO-deficient virus (Figure 3D and Appendix A). The behavior of ∆M36 affirmed previous findings [29] showing that vICA was necessary to suppress TNFR1-dependent cell death (Figure 3A). M45*mut*RHIM-infected WT or *Tnfr1^−/−^* cells exhibited similar susceptibility to cell death that remained dependent on necroptosis mediator RIPK3, but not on RIPK1, CASP8, nucleic acid sensor ZBP1, or BAX and BAK (Figure 3B–D and Appendix A). Interestingly, *Bax^−/−^Bak^−/−^* cells exhibited enhanced susceptibility to the M45*mut*RHIM-dependent necroptosis. These observations clearly lay out the key host targets for each cell death suppression mechanism. M45-encoded vIRA specifically suppresses RIPK3-dependent signaling independently of TNF, CASP8, or BAX/BAK; M36-encoded vICA suppresses TNF-dependent CASP8 activation independently of RIPK3, BAX and BAK; M38.5/M41.1-encoded vMIA/vIBO specifically suppresses BAX- and BAK-dependent signaling where TNF appears to contribute to earlier events. Curiously, M45*mut*RHIM-infected *Bax^−/−^Bak^−/−^* cells exhibited enhancement when compared with WT settings (Appendix A), suggesting that even though the mitochondrial pro-death proteins are overall dispensable, they restrain necroptotic signaling. Thus, in macrophages, MCMV employs distinct, non-redundant suppression strategies to prevent infection-induced necroptosis, apoptosis, and mitochondrial cell death. All PCD suppressors are necessary to maintain viral replication, as well as full pathogenesis in vivo (Figure 1B,C and [30]).

### 3.5. vMIA and vIBO Interface with vIRA and vICA

To evaluate the temporal aspects of PCD suppression by vIRA, vICA, vMIA and vIBO, we compared the parental virus to M45*mut*RHIM (inducer of RIPK3-dependent necroptosis), ∆M36 (inducer of CASP8-dependent apoptosis), and ∆M38.5/M41.1 viruses during infection of WT macrophages (Figure 3E and Appendix A). Cell death was measured by integration of specific fluorescent signal of cell-impermeable dye Sytox Green as the execution of PCD signaling increases membrane permeability. Neither mock treatment nor K181 infection induced permeability. M45*mut*RHIM-infected cells exhibited a rapid increase in cell permeability within 6 to 9 hpi. Necroptosis was evident by 12 hpi (Figure 3E and Appendix A, third row). ∆M36 and ∆M38.5/M41.1 mutant viruses revealed comparable timing of cell death, first becoming evident by 12 to 15 hpi (Figure 3E). ∆M38.5/M41.1-induced cell death remained overall less prominent compared with that induced by ∆M36 infection (Appendix A fourth and fifth rows). This relatively slow development of PCD with vMIA/vIBO double-deficient viruses was expected for infection-triggered mitochondrial stress as observed with HCMV vMIA mutant or MCMV single mutant disrupting M38.5 or M41.1 expression [13,17,31,36,38]. Therefore, M45*mut*RHIM-induced necroptosis represents the most robust and immediate form of cell death in MCMV-infected BMDM, followed by ∆M36-induced extrinsic apoptosis and ∆M38.5/M41.1-induced mitochondrial cell death. Elaboration of PCD suppression during MCMV infection of macrophages controls this temporal network. Each suppressor functions independently of the others with a feed-forward crosstalk between pathways as indicated by the contributions of TNF to the mitochondrial signaling (Figure 3A,B) or BAX and BAK to necroptotic signaling (Appendix A), as well as the role of CASP8 suppression in the induction of necroptosis [30].

### 3.6. vIRA, vICA, vMIA and vIBO Interface with Innate Inflammation

Cytokines induced during infection synergize with PCD signaling (mitochondrial and non-mitochondrial) in cells and tissues [75,100,101]. We sought to assess the potential for communication between PCD pathways, virus-encoded death suppressors, and innate inflammation. Analysis of cytokine and chemokine release from WT BMDM showed that infected M45*mut*RHIM-, ∆M36-, and ∆M38.5/M41.1-infected cells show patterns of inflammation that are distinct from K181 as well as from each other (Figure 3F and Appendix A). M45*mut*RHIM induced ~70% loss of viability (Figure 3B) and a reduction in cytokines and chemokines except for IL-27 (Figure 3F and Appendix A). ∆M36 induced ~50–60% viability loss associated with a surprising elevation of several cytokines and chemokines including IL-27, CXCL-10 (IP-10), CXCL-1 (KC), M-CSF, Mip-1α, and MIP-1β. Several others including siCAM-1, IL1ra, IL-10, IL-16, and MCP-1 (JE) did not show a noticeable difference from the K181 infection setting. ∆M36-induced G-CSF, IL-6, MCP-5, MIP-2, RANTES, SDF-1, TIMP-1, and TNF were reduced compared with K181. ∆M38.5/M41.1 induced ~30–40% viability loss but a dramatic reduction in overall cytokines and chemokines. Thus, during the course of infection, macrophages may support necroptosis with reduced inflammatory markers, extrinsic apoptosis with elevated inflammatory markers, and activated BAX/BAK-dependent signaling with reduced inflammatory markers. From these observations, it appears that vIRA, vMIA, and vIBO potentiate innate inflammation by restraining the host factors RIPK3, BAX, and BAK, whereas vICA suppresses inflammation by preventing CASP8 activation. Overall, concerted functions of vIRA, vICA, vMIA, and vIBO dictate optimal infection conditions for MCMV by regulating death signaling and replication, as well as innate inflammation.

### 3.7. Ripoptosome Components Are Dispensable in vMIA/vIBO-Dependent Signaling

To identify crosstalk between extrinsic and intrinsic cell death components during MCMV infection, we assessed cytosolic death players RIPK1 and CASP8 because of their key roles in PCD signaling [102,103,104,105,106]. Extracellular or intracellular stress often induces the association of adaptor FADD with RIPK1 and CASP8, leading to the formation a multi-functional, pro-death platform known as the ripoptosome [106,107,108,109]. RIPK1 engagement has been observed downstream to both extrinsic TNF-dependent and intrinsic BAX/BAK-mediated signaling [110,111,112]. MCMV infection engages RIPK1, CASP8, and RIPK3 in BMDM [29,30], but how these proteins play out in the context of vMIA/vIBO-dependent signaling remains unresolved. RIPK3 and ZBP1 may contribute to ripoptosome function. We infected WT, *Ripk3^−/−^*, *Casp8^−/−^Ripk3^−/−^*, *Ripk1^−/−^Casp8^−/−^Ripk3^−/−^*, and *Zbp1^−/−^* BMDM, as well as cells expressing kinase inactive versions of RIPK1 (*Ripk1^K45A/K45A^*) or RIPK3 (*Ripk3^K50A/K50A^*) with K181 or ∆M38.5/M41.1 mutant virus (Figure 4A,B and Appendix A). All genotypes exhibited similar susceptibility to vMIA/vIBO-deficient virus-induced cell death (Figure 4A,B and Appendix A) establishing that RIPK1, RIPK3, CASP8, and ZBP1 were completely dispensable for mitochondrial cell death. Immunoblot analysis of WT, and *Ripk1^−/−^Casp8^−/−^Ripk3^−/−^* BMDM infected with K181, M45*mut*RHIM, ∆M36, or ∆M38.5/M41.1 established activation patterns of caspase implicated in extrinsic apoptosis (CASP8-CASP3-CASP7) [29] or mitochondrial cell death (CASP9-CASP3-CASP7) [113,114] at 24 hpi (Figure 4C). WT, but not *Ripk1^−/−^Casp8^−/−^Ripk3^−/−^* cells infected with K181 or M45*mut*RHIM, differed from mock infected cells by a detectable increase in levels of cleaved CASP7 (~19 and 17 kDa). Thus, MCMV infection activates CASP7 independently of vIRA, vICA, and vMIA/vIBO, but is dependent on ripoptosome components. However, this activated CASP7 did not result in the death of WT cells during infection with K181. Infection with ∆M36 resulted in the expected appearance of cleavage-activated CASP8 (~43 and 18 kDa), CASP7 (17 kDa), and CASP3 (17 kDa) cleavage products. Activated CASP9 (~39–37 kDa) levels were also detected, indicating that this mitochondrial cell-death-associated caspase was engaged during ∆M36 infection, even though mitochondrial BCL2-family proteins were dispensable for the death [29]. These data clearly showed that removal of vICA-dependent CASP8 suppression has a cascade impact on activation of multiple caspases beyond CASP8. As expected, *Ripk1^−/−^Casp8^−/−^Ripk3^−/−^* cells that remained completely resistant to ∆M36-induced apoptosis [30] resisted the activation of CASP8 and CASP3, but surprisingly continued to exhibit activation of CASP9 and CASP7. It remains surprising that *Ripk1^−/−^Casp8^−/−^Ripk3^−/−^* cells showed full activation of proapoptotic caspases despite the absence of core ripoptosome components. However, CASP9/CASP7 activation in these cells did not drive death in the absence of these critical ripoptosome components. With ∆M38.5/M41.1 infection, WT cells exhibited activation of CASP9, CASP7, and CASP3. CASP8 was activated over K181 but in reduced levels compared to ∆M36, consistent with the requirement for vICA in CASP8 suppression. *Ripk1^−/−^Casp8^−/−^Ripk3^−/−^* cells also activated CASP9, CASP7, and CASP3, but at levels modestly reduced compared with WT cells. Death did not differ between the genotypes (Figure 4A,B), demonstrating that the lower levels of the CASP-activation pattern expressed in *Ripk1^−/−^Casp8^−/−^Ripk3^−/−^* cells were sufficient to induce mitochondria-dependent death signaling. The lower levels of CASP8 activation may reflect the contribution from TNFR1 (Figure 3B), which is known to signal through RIPK1 and CASP8 [109]. Therefore, extrinsic death signaling does not impact double mutant-induced mitochondrial death signaling. vMIA/vIBO double-deficient virus maintained attenuation in *Casp8^−/−^Ripk3^−/−^* BMDM (Appendix A–C). In salivary glands from *Casp8^+/−^Ripk3^−/−^* mice, this attenuation was evident at 14 dpi (Appendix A). However, in *Casp8^−/−^Ripk3^−/−^* mice, double-deficient MCMV exhibited a spread in viral titer, indicating TNF-CASP8-dependent signaling may amplify the innate immune response to restrict virus infection in vivo. Therefore, vMIA and vIBO target the mitochondria to prevent cell death and benefit fitness independently of known extrinsic PCD signaling components and pathways. Other PCD signaling pathways potentially influence outcomes in vivo.

In summary, we established that the mitochondria are central sensors of MCMV infection via multiple processes. Mitochondrial serine protease HtrA2/Omi restrains replication (Figure 5) while HtrA2/Omi-independent signaling engages PCD signaling. M38.5/M41.1-encoded vMIA and vIBO suppress mitochondrial signaling by: (a) modestly interfering with HtrA2/Omi and (b) preventing cell death through suppression of BAX/BAK-dependent signaling. Overall, MCMV suppresses PCD in the follow ways: M45 (vIRA) suppresses RIPK3-dependent necroptosis, with ZBP1 contributing in non-myeloid cells but not in macrophages; M36 (vICA) suppresses the TNF signaling-dependent CASP8-CASP3 activation axis, as well as the CASP8-independent cleavage-activation of CASP9 and CASP3; vMIA and vIBO together suppress mitochondrial signaling. vIRA, vICA, or vMIA/vIBO target distinct host proteins blocking non-redundant PCD pathways. Our data reveal crosstalk between different PCD pathways in macrophages. TNFR1 contributes to the mitochondrial cell death observed in absence of vMIA and vIBO; BAX and BAK restrained necroptosis in the absence of vIRA, suggesting cell death initiated by mutant viruses benefits from host PCD proteins involved in other forms of cell death as a feed-forward loop. Therefore, MCMV orchestrates an efficient arsenal of sequential, independent, and non-overlapping suppression strategies against the interconnected mammalian PCD benefitting fitness during natural infection of cells or intact hosts.

## 4. Discussion

Here, we reveal an intricate interplay between virus-encoded PCD suppressors, mammalian PCD machinery, and TNF-dependent signaling. Cellular pathways may collaborate to undermine cell viability, cut short virus replication, and drive infection-associated inflammation in the absence of dedicated MCMV-encoded PCD suppressors. We introduce new concepts regarding the involvement of mitochondria in this infected cell death-inflammation matrix. Mitochondrial HtrA2/Omi restrains MCMV replication, reminiscent of its role in restricting HCMV infection in fibroblasts via the induction of PCD [13,49]. However, in MCMV-infected BMDM, this serine protease limits infection independently of PCD. Recently, it has been demonstrated that HCMV vMIA suppresses BAX/BAK-dependent signaling to facilitate the infection of mast cells [37]. Future investigations with HCMV and MCMV in myeloid and non-myeloid lineages will be necessary to resolve potential differences between HCMV studies and our current observations with MCMV. Tissue-specific contributions from HtrA2/Omi or differences between the human and murine viruses might certainly be expected given the ninety million years of evolutionary divergence in these two biologically similar viruses. Overall, we add MCMV vMIA/vIBO as a modest suppressor of HtrA2/Omi to the known herpesvirus-encoded suppressors, HCMV-encoded vMIA [13] and HHV-8-encoded vIRF1 [115]. Studies with MCMV single mutant for either vMIA or vIBO have established BAX and BAK as specific targets for the viral suppressors [31,38]. We expanded that understanding by implicating HtrA2/Omi and TNF-dependent signaling as additional targets. vMIA/vIBO double-deficient MCMV failed to efficiently express the immediate early viral protein IE1 in macrophages (Appendix A). The other cell death suppressor, mutant MCMV generated on this K181-BAC parental background, as well as single mutants for vMIA or vIBO did not show a similar defect [30,31,32], indicating vMIA/vIBO may potentially regulate the initial steps controlling entry prior to immediate early gene expression. Future studies in primary and immortalized murine cells will be necessary to determine the full scope of contribution by these suppressors during infection. During infection, vMIA/vIBO suppress an early involvement of TNF beyond BAX/BAK such that the double-mutant virus drives TNFR1-dependent death by 24 hpi (Figure 3B). In *Casp8^−/−^Ripk3^−/−^* but not *Casp8^+/−^Ripk3^−/−^* mice, vMIA/vIBO mutant MCMV exhibited a spread in replication (Appendix A). TNF signaling showed an intricate association with both HCMV and MCMV infection [12,13,17,18,29,75,116,117,118], although never before implicated in mitochondrial signaling. Our observations indicated that TNF-dependent signaling augments vMIA/vIBO mutant MCMV-driven death in macrophages and amplifies response in mice via CASP8 activation.

Surprisingly, TNF-driven death during vMIA/vIBO-deficient mutant infection occurs independently of CASP8 and despite the presence of virus-encoded vICA. Thus, vICA successfully restricts the known TNFR1-CASP8 death signaling. TNFR1 plays a role early in infection, but BAX/BAK activity bypasses any contribution from this cytokine later in infection. *Bax^−/−^Bak^−/−^* BMDM exhibit a partial rescue from double mutant-induced death. Elimination of TNFR1-dependent signaling along with BAX/BAK must be evaluated for impact on cellular viability and in vivo replication during double-mutant virus infection. From the data shown here, it appears that overlapping intrinsic and extrinsic PCD signaling pathways may converge in the mitochondria late during MCMV infection where vMIA/vIBO mitigates impacts from both pathways. BAX/BAK-interacting partners BID, BIM, and PUMA are individually dispensable but contribute together, consistent with the potential diversity of BAX and BAK activators as infection progresses. Whether HCMV vMIA elicits a similar suppression strategy against both extrinsic and intrinsic death signaling will come from future assessment. TNF is known to induce extrinsic death via cytosolic FADD-RIPK1-CASP8 association [119], with mitochondrial amplification in BID-dependent as well as BID-independent signaling [57,58,120,121,122,123]. RIPK1 and CASP8 are elevated by 14 hpi in infected macrophages [29] and possibly contribute to the early TNFR1 impact but remain ultimately uninvolved in the pathway suppressed by vMIA/vIBO. The role of BID, the recognized bridge between TNF and mitochondrial amplification [44,57,58,120,122,123], remains unclear. The picture that emerges regarding vMIA/vIBO suppression seems reminiscent of adenovirus-encoded viral BCL-2 antagonist E1B 19K, a function that suppresses analogous TNF-mitochondria synergy during infection [124]. There may be a common pattern during DNA virus infections. Our observations demand future experiments to elucidate how the TNF-mitochondria synergy plays out in cells as well as in vivo, especially during MCMV pathogenesis settings where TNF signaling contributes to outcomes [74,75,125].

Even though the host players for extrinsic apoptosis, necroptosis and mitochondrial death, exhibit limited crosstalk, a remarkable partitioning exists between the pathways MCMV-encoded PCD suppressors vIRA, vICA, and vMIA/vIBO hold under control. MCMV has evolved to antagonize amplification of the antiviral response as infection progresses such that each suppressor functions non-redundantly in particular subcellular location. Immediately following the infection of macrophages, MCMV faces RIPK3 in the cytoplasm. vIRA, a tegument-associated protein, is expressed early to usurp RIPK3 RHIM engagement [17,126]. vIRA also disrupt RIPK1, ZBP1, and TRIF RHIM-dependent signaling [14,72,82]. However, all these players, as well as TNF and types I and II interferons, remain dispensable for vIRA function during MCMV infection of macrophages (Figure 3A and Appendix A and [30]). In fibroblasts or endothelial cells, the nucleic acid sensor ZBP1 initiates this signaling [14,15,30]. Therefore, even though vIRA-dependent RIPK3 suppression is critical for MCMV infection and pathogenesis independent of cell type [30], the requirements to trigger this signaling in macrophages are less stringent than in non-myeloid cells. Restricting virus in macrophages may be of critical importance for the host such that additional requirements beyond RIPK3-MLKL are bypassed. BAX/BAK surprisingly restrain necroptosis unleashed in the absence of vIRA such that vIRA-deficient MCMV drives enhanced death in BAX/BAK-deficient cells (Appendix A). This feedback loop in combination with our identified TNF-mitochondria crosstalk reveal that extrinsic apoptosis, necroptosis, and mitochondrial cell death overlap during MCMV infection. The suppressors function in a non-overlapping fashion to subvert the integrated network of host PCD pathways. vIRA functions are followed by requirements for vICA in cytoplasm and vMIA/vIBO in mitochondria as host PCD pathways become activated. Pyroptosis remains dispensable for all these processes (Appendix A and [29]). Thus, MCMV subverts distinct PCD pathways by sequential activation of PCD suppressors that target distinct PCD pathways in different cellular localizations.

The suppressor-PCD interface during MCMV infection influences innate inflammation such that by 24 hpi, distinct patterns of cytokine/chemokine release became evident in the absence cell death suppression (Figure 3F and Appendix A). vIRA-deficient virus exhibited an overall hypo-inflammatory phenotype. In mice, this mutant virus associates with elevated innate and adaptive responses [30]. MCMV-infected cells undergoing necroptosis may be inherently inflammatory by releasing danger signals, but it remains to be determined whether the balance of signaling undermines viability or protects surrounding cells. Remarkably, vICA-deficient virus infection also exhibited a hyper-inflammatory phenotype. This could be due to either of two reasons. This apoptotic signal may itself be inflammatory. By suppressing CASP8, vICA blocks both death-dependent as well as death-independent inflammation that appear to be unleashed during ∆M36 infection. We previously suggested this possibility because vICA-deficient virus infection induced higher levels of TNF early during infection [29]. At the time of assessment (24 hpi), vMIA/vIBO double-deficient virus infection produced a hypo-inflammatory state, suggesting that unleashed BAX and BAK reduce the amplification of inflammatory signaling. Thus vMIA/vIBO appear to be potentiators of inflammation late during MCMV infection, presumably because this benefits the virus in some way. The consequences of each suppression mechanism on pathogenesis require further assessment.

In summary, we showed that PCD pathways restrict MCMV infection in macrophages. Sequential engagement of vIRA, vICA, and vMIA/vIBO, suppressors of RIPK3-dependent necroptosis, TNFR1-CASP8-dependent apoptosis, and mitochondrial signaling, respectively, regulate levels of inflammation in cultured macrophages and *in vivo*. Combined functions of all suppressors establish a protective shield for CMV infection, thereby establishing these suppressor pathways as potential therapeutic targets during viral pathogenesis. Deactivation of one or more PCD suppressors will restrict virus infection. Elimination of the mitochondrial suppressors (vMIA/vIBO) reduces viral fitness and overall inflammation. Given that the functions of these suppressors are conserved between MCMV and HCMV, we reveal that targeting the mitochondrial suppressors of CMV may be a safe, viable way to target this virus.

## Figures and Tables

**Figure 1 viruses-13-01707-f001:**
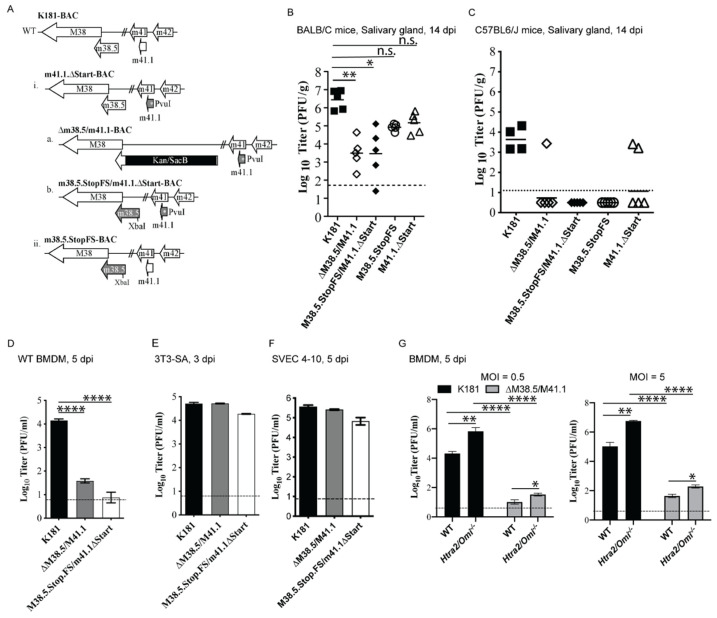
∆M38.5/M41.1 mutant MCMV is attenuated for replication in vivo in macrophages. (**A**) Diagrammatic representation of the mutations introduced into M38.5 (vMIA) and M41.1 (vIBO). (**B**,**C**) Titer of parental MCMV (K181) or indicated mutant viruses in salivary glands from 6-week-old BALB/c (**B**) or C57BL6/J (**C**) mice at 14 dpi following inoculation with 2 × 10^5^ PFU intraperitoneally (**B**) or 5 × 10^6^ PFU subcutaneously into a footpad (**C**). Each symbol represents the titer obtained from a single mouse. Datasets are compared using one-way ANOVA and the mean for each dataset is indicated. (**D**–**F**) Replication in C57BL6/J wild type (WT) murine BMDM infected at multiplicity of infection (MOI) of 0.5 (**D**); 3T3-SA fibroblasts (**E**) and SVEC4-10 endothelial cells (**F**) infected at MOI of 0.3. Data shown are pooled results from three experiments for each setting and the dotted line indicates the limit of detection for each group. Each data set is compared using one-way ANOVA. (**G**) Replication of K181 or ∆M38.5/M41.1 virus in WT and *Htra2/Omi^−/−^* BMDM at 5 dpi following infection at an MOI = 0.5 (left panel) or 5 (right panel). For each panel, data shown are pooled results from at least three experiments. Horizontal lines indicate data compared by Wilcoxon matched-pairs signed rank test. For panels (**D**–**G**), error bars indicate the standard error and mean for each dataset. * is *p* < 0.05; ** is *p* < 0.01; **** is *p* < 0.0001, n.s. is non-significant.

**Figure 2 viruses-13-01707-f002:**
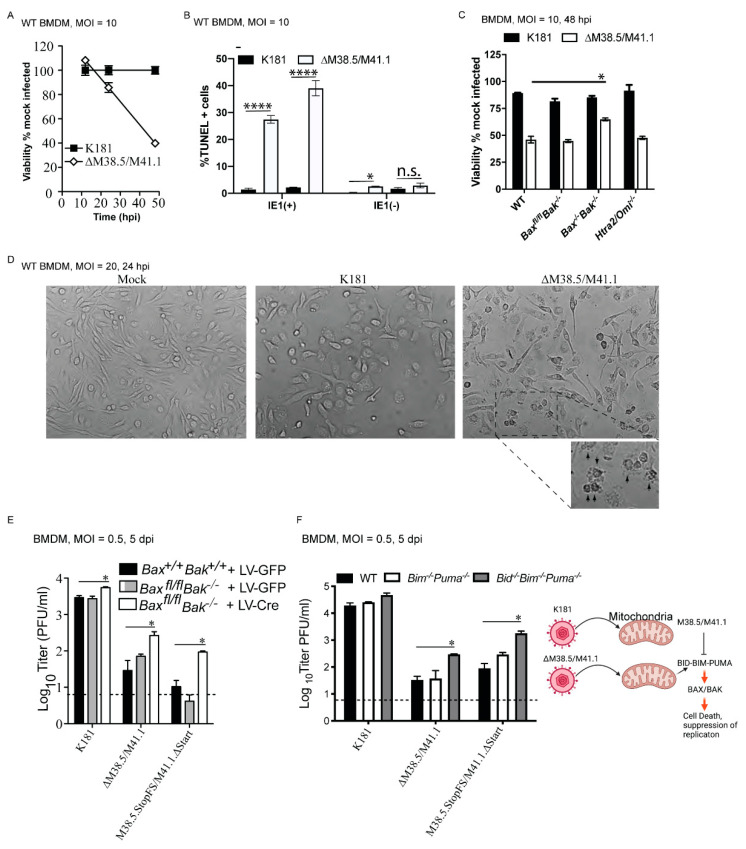
M38.5 and M41.1 suppress BAX- and BAK-dependent BMDM viability. (**A**) Viability of WT BMDM infected with K181 or ∆M38.5/M41.1 (MOI = 10) measured by the loss of ATP at indicated times. Each bar or symbol represents pooled data obtained from a minimum of three replicates. (**B**) Quantification of DNA fragmentation by TUNEL assay in BMDM (MOI = 10) that either show (+) or lack (−) viral antigen IE1 at 24 and 48 hpi. Each bar represents pooled data from three replicates and shows the average percentage of TUNEL positive cells. (**C**) Viability of WT, *Bak^−/−^Bax^fl/fl^*, *Bax^−/−^Bak^−/−^*, as well as *Htra2/Omi^−/−^* BMDM under the conditions described in 2A. Comparison between ∆M38.5/M41.1 infection in WT and *Bax^−/−^Bak^−/−^* cells was performed using Wilcoxon matched-pairs signed rank test. (**D**) Cell morphology during infection was visualized by phase-contrast microscopy of infected BMDM (MOI = 20). Representative images were obtained at 24 hpi. The highlighted area in the right-most panel exhibits visible apoptotic cell morphology (indicated by black arrowheads). (**E**) Replication of K181, ∆M38.5/M41.1, or M38.5.StopFS/M41.1∆Start virus in *Bax^+/+^Bak^+/+^* and *Bax^fl/fl^Bak^−/−^* BMDM previously transduced with a lentivirus expressing GFP (LV-GFP) or Cre recombinase (LV-Cre) at 5 dpi (MOI = 0.5) with MCMV. (**F**) Replication of K181, ∆M38.5/M41.1, or M38.5.StopFS/M41.1∆Start virus in *Bim^−/−^Puma^−/−^* and *Bid^−/−^Bim^−/−^Puma^−/−^* BMDM at 5 dpi (MOI = 0.05). For each experiment, data shown are pooled results from at least three experiments. Horizontal lines indicate groups of data being compared by Wilcoxon matched-pairs signed rank test. Comparisons of the three bars shown in Panels E and F were performed using one-way ANOVA. Error bars indicate standard error and mean for each dataset. * is *p* <= 0.05; **** is *p* < 0.0001; n.s. is non-significant. Cartoon diagram (**F** right panel) generated using BioRender illustrates the function of M38.5/M41.1 during MCMV infection of macrophages.

**Figure 3 viruses-13-01707-f003:**
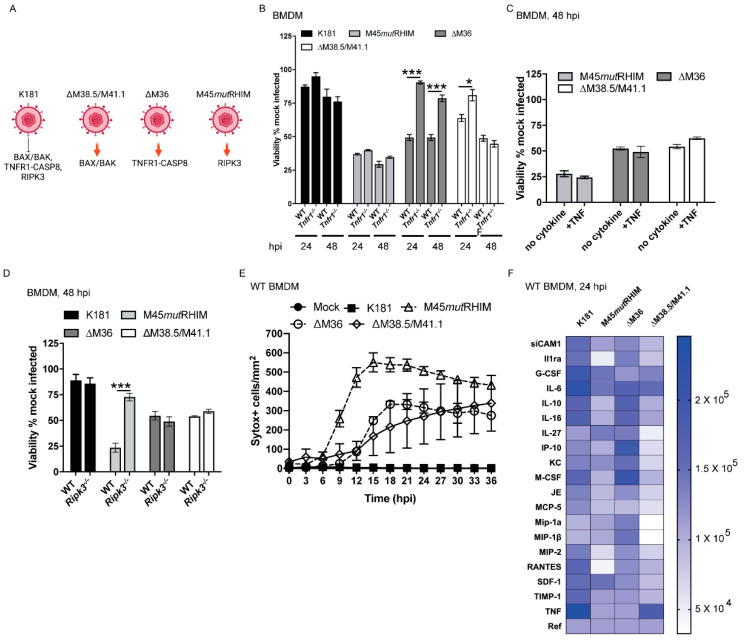
Characterization of MCMV-dependent suppression of apoptosis and necroptosis. (**A**) Pictorial representation generated using BioRender software depicting the cell death suppressor pathways recognized to function in MCMV-infected macrophages. (**B**–**D**) Viability measured at indicated times post infection with K181 or ∆M38.5/M41.1 (MOI = 10) in WT and *Tnfr1^−/−^* BMDM (**B**), or WT BMDM treated with TNF (25 ng/mL) 1 hpi (**C**), or WT and *Ripk3^−/−^* BMDM (**D**). Each bar represents pooled data obtained from minimum of three experiments. Horizontal lines indicate the two groups being compared by Wilcoxon matched-pairs signed rank test. * is *p* < 0.05; *** is *p* < 0.001. Only significant differences are indicated by horizontal lines in panels. (**E**) Viability of WT BMDM infected with K181, ∆M36, M45mutRHIM, or ∆M38.5/M41.1 MCMV assessed by inclusion of cell-permeable dye (Sytox Green) over the indicated timecourse. Each graph represents the pooled data from two independent experiments. Error bars indicate the standard error and mean for each dataset. (**F**) Heat map depicting a broad survey of cytokines and chemokines detected in cell-free supernatant from WT BMDM infected with the indicated viruses for 24 h. Each group contains pooled supernatants from two independent experiments. Scale bar on the right indicates range intensity of the cytokine signal. Reference (Ref) is generated from average of three reference points located on dot blot (see Appendix A). For each mutant infection setting, references are normalized with respect to K181 infection.

**Figure 4 viruses-13-01707-f004:**
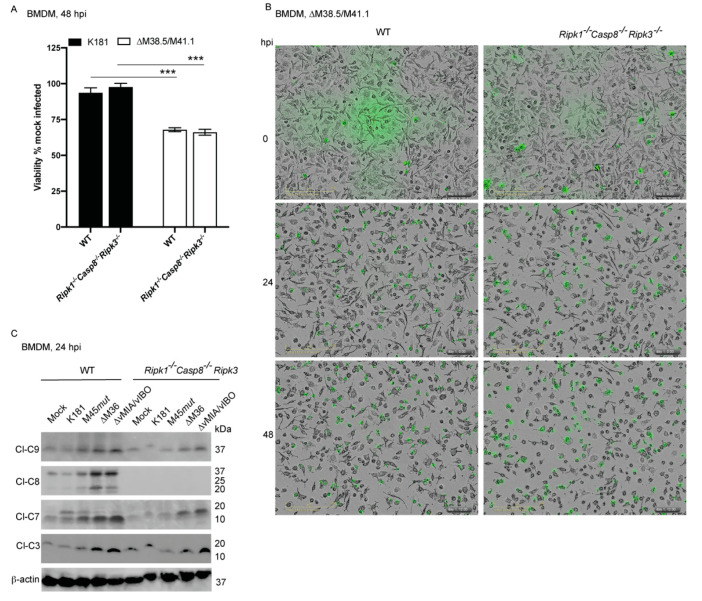
Ripoptosome and mitochondrial stress. (**A**) Viability of WT or *Ripk1^−/−^Casp8^−/−^Ripk3^−/−^* BMDM infected with K181 or ∆M38.5/M41.1 (MOI = 10) assessed by the loss of ATP at 48 hpi. Each bar represents the pooled data from at least three experiments. Horizontal lines indicate the two groups being compared by Wilcoxon matched-pairs signed rank test. Only significant relevant comparisons are indicated in panels. Error bars indicate standard error and mean for each dataset. *** is *p* < 0.001. (**B**) Representative light microscopy images (lower left scale bar = 200 mM) of WT or *Ripk1^−/−^Casp8^−/−^Ripk3^−/−^* BMDM infected with indicated viruses at indicated times. Time stamp from the microscope is included in the lower-right corner of each image. (**C**) Immunoblot assessment to detect the appearance of cleaved forms of CASP9 (Cl-C9; ~37 kDa), CASP8 (Cl-C8; 43 and 18 kDa), CASP7 (Cl-C7; 18 kDa), and CASP3 (Cl-C3; 17 kDa) in cellular protein lysates from *Casp8^−/−^Ripk3^−/−^* BMDM treated with medium (mock) or infected with the indicated viruses. β-actin (38.5 kDa) serves as the loading control.

**Figure 5 viruses-13-01707-f005:**
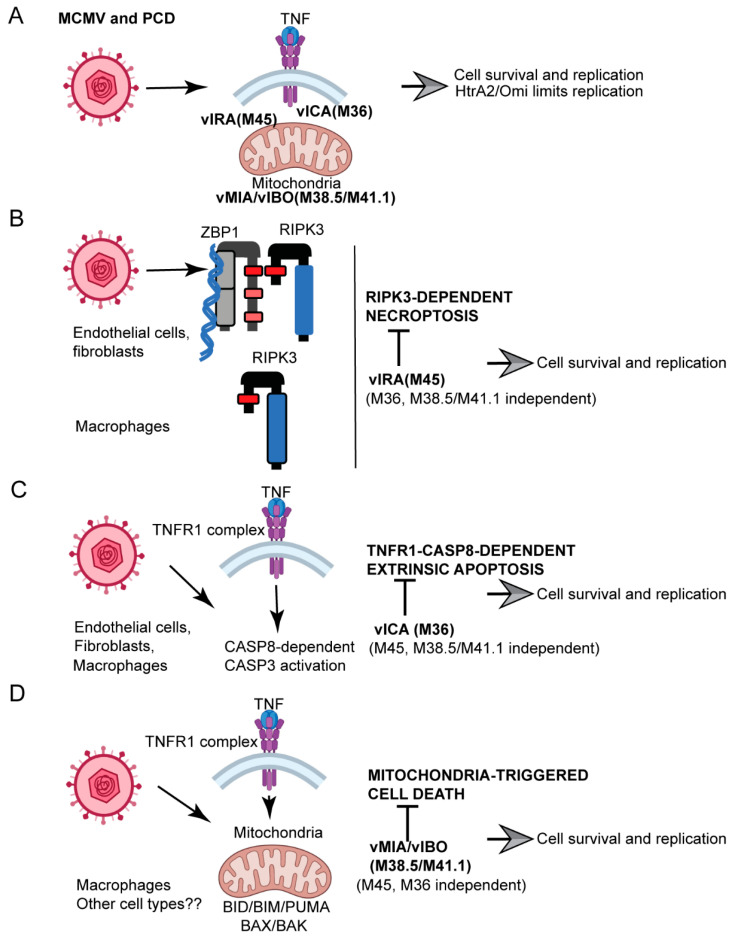
Schematic summary of how PCD pathways interface with MCMV infection. (**A**) HtrA2/Omi regulates MCMV replication in macrophages. (**B**) vIRA (M45) suppresses RIPK3-dependent necroptosis via ZBP1-dependent (in nonmyeloid cells) and ZBP1-independent (in macrophages) pathways. (**C**) vICA (M36) suppresses TNFR1-dependent activation of CASP8 in myeloid and non-myeloid cells. (**D**) vMIA/vIBO (M38.5/M41.1) suppresses BID/BIM/PUMA-mediated BAX/BAK-dependent mitochondrial death signaling. Diagram was generated using BioRender Software.

## Data Availability

All data are included in main manuscript and supplemental information. Raw data are available upon request from P.M. and E.S.M.

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
