# Peer review of "Multiple Autonomous Cell Death Suppression Strategies Ensure Cytomegalovirus Fitness"

_viruses, 2021, doi:10.3390/v13091707_

Round 1

Reviewer 1 Report

The manuscript by Mandal and colleagues expands our understanding of the regulation of cell death pathways by MCMV. The authors analyzed M38.5/M41 (vMIA/vIBO) deletion mutants to show that combined BAX/BAK suppression is essential for virus growth in macrophages and the successful viral spread to the salivary glands in vivo. Further, the authors analyzed the timely coordination of the inhibition of the different death pathways during MCMV infection and the influence of the viral counterplayer on the induction of inflammation signals in macrophages. Whereas the scientific value of the investigation is fine and worth publishing in Viruses, the manuscript needs an enhancement before acceptance.

Major points:

  1. My major concern about the submitted version is the, apparently, lack of proofreading prior to submission. In the following, I list all problems that I was aware of during reading, but I am sure that the list is not complete. Therefore, the authors have to check all Figures and figure references throughout the manuscript.
  • Figure 4 is not placed correctly in the manuscript.
  • Line 434 Figure 1F and its legend are missing
  • Line 358 wrong figure reference, this cannot be Figure S1B
  • Line 360 wrong figure reference, this should be Figure 2c
  • Whole Figure 2, the figure and its legend do not fit together, apparently, Fig 2B and 2c have to be changed.
  • Line 499 figure references do not fit, it cannot be S3A-C
  • Line 618, this should be S3A, not S3B
  • In Figure S3C (DM36) the pictures seem to mix up because the green disappears and then reappears.
  1. In section 3.1, the authors discuss viral fitness in vivo, but only showed a single time point (14dpi) of a single organ (SG) in two mouse strains. Here a reduced viral titer is visible, but it remains unclear what is the reason for this observation. Is it due to limited dissemination to the SG or due to a reduction in virus growth because of enhanced apoptosis? Here I miss the discussion of the author's own work and by others (Farrell, Reddehase, Adler; PMID:34132572, PMID: 31092580, PMID: 28974616, PMID: 24629341, PMID: 10485920, PMID: 8083964, PMID: 25782576, PMID: 23935483). Do the authors also analyzed other organs and earlier time points, this might be included to see if the phenotype is mainly a spreading or replication phenomenon. In the same line the statement in lines 293-295, should be changed, dissemination phenotype is not similar to viral fitness for a discussion see PMID: 25782576.

In addition, the authors showed that the double mutant has a growth phenotype in BMDM but not in ECs and fibroblast. But here the authors compare two transformed cell lines (large T antigen-positive) with non-transformed macrophages. Do the authors check if the transformation influences the induction of BAX/BAK and therefore the growth of the double mutant? They might infect MEF as a surrogate for a non-transformed non-macrophage cell.

  1. In Figure 4C, the loading control is overexposed, and therefore comparable protein amounts are hard to see. This is a problem because the authors discuss a reduction in protein amounts. Perhaps here a quantification with normalization might be helpful. It seems that the authors used the same blot after stripping to detect the specific Cl-Cs but this is not mentioned. Otherwise, a second loading control is needed.

Minor points

  1. It is unclear how the authors calculated the t-test. It is a good choice to use the Welsch-correction, but they should mention if they used the log-transformed data which is correct, as virus titers are log-normal distributed) of the non-transformed data (which is incorrect, due to the log-normal distribution of the data). In Figure S1B, the t-test cannot be used, because most of the data are below the detection limit, I recommend the use of a non-parametric test for analysis. Same for S4D, here no statistical test is mentioned.
  2. In Figure S1A please include the virus name in the Figure which makes it more understandable
  3. In Figure 3, I would recommend including the original dot blots of the cytokine array, which gives an additional visualization of the results
  4. Throughout the manuscript, I miss the number of measurements included in the bar diagrams, also the description of the error bars should be included
  5. I recommend marking the median not the mean in Figures 1B, S1B, S4D.
  6. Please add the explanation for the green dye in figure legend 4 and S3, I needed some time to understand it.
  7. Reference 68 is missing
  8. For me Reference 74 do not fit optimal, perhaps PMID: 31011793 suits better
  9. Lines 564-565 it is unclear which function has been confirmed from this study
  10. Line 18 for me cytomegaloviruses makes more sense than cytomegalovirus
  11. Paragraph 2.4 it confusing to mention first the cell numbers and then the assay
  12. Line 622. The statement is too strong as Sacher et al. (PMID: 22114552, PMID: 18407069) showed that macrophages have only limited contribution on the total virus progeny and virus line dissemination.
  13. Line 639 the first “cells” is not needed
  14. Some fonts are very small and hard to read e.g. Figure 4A
  15. The authors might use Figure 4D as a separate Figure 5 to summarize the PCD pathways

Reviewer 2 Report

Mocarski and colleagues are world leaders in this area of research and here provide a new insight into the regulation of viral replication by cell death - and the outcomes of measures employed by the virus to subvert them. 

Figure 1 investigates the replication of the viruses in vivo and demonstrates the importance macrophages in the context of this study as the data clearly show a macrophage-dependent phenotype for replication. It is logically inferred this will reflect as a fitness defect through reduced transmission presumably due to lower titres of virus in the salivary tissue. I assume this has been demonstrated in MCMV transmission studies (that titre is the direct correlate of transmission) and thus could be cited?

Were neuronal cells investigated - i.e. is the wholly macrophage specific? This may be important in the context of pathogenesis and neuronal cells have been shown to have very similar sensitivities to cell death pathways as macrophages

Figure 2 moves to Htra/Omi mediated regulation as they have previously shown this is important for HCMV infection via an interaction with vMIA. The data show that IE1 positive cells are less likely to be TUNEL +ve in wt virus indicating a clear survival phenotype. I assume in IE1 neg cells there is no difference in TUNEL staining in the populations infected with the two viruses?

That said the figure legend relates an analysis of wt as well as Bax/bak KO mice yet I cannot identify the data clearly in the figure itself as labelled in legend (i assume the mean Fig 2C). The manuscript does seem a little dis-organised in general (figure 4 appears before figs 1-3 in the manuscript file for example). 

The data demonstrate the defect in the deletion virus is not rescued by deletion of cell death proteins Bak, Bax, and omi except a double deletion in Bax/Bak where the Bak is floxed where a small restoration is seen - this could be explained better as it is a subtle technological difference that is could be confusing.

Here the supplementary data could be included in the main manuscript - the data that demonstrates the 3 KO of the 3 BH3 proteins is highlighting the complexity of the defense mechanisms - the host has multiple ways to kill which the virus has to deal with. It seems to me this should be in the main manuscript

Figure 3 is the key experiment regarding timing effects and demonstrates that early on in infection TNF is important. Here the authors use previous viruses shown to modulate TNFR signalling and a TNF independent virus to demonstrate specificity.  The double mutant is clearly sensitive to TNFRI signalling - it is also notable that this virus is highly inflammatory for TNF production (like wt). Do the single mutants used in Fig.1 and 2 show reduced TNF production like the single mutants used here in fig 3? Could this also contribute to the cell death phenotype seen with single and double mutants in Figs 1 & 2. For example, if you challenge infected with macrophages with recombinant TNF is there differential sensitivity of the single and double mutant viruses?

Figure 4 (which is the first figure in the manuscript) - in A the authors measure ATP to establish that MCMV lacking cell death suppressors are less viable. Although there is a reduction (which is significant) there remains substantial ATP activity. How does this phenotype compare to other insults of the same pathways for comparison?

Minor comments below:

Ln121 - this last paragraph needs tweaking. 'Otherwise' what? as written it isnt easy to follow the logic of the authors.

In the opening paragraph when the importance of PCD is discussed in the context of infection could a role in species specificity (a form of restriction) be cited?

Figure 4 appears in the manuscript file before Figs 1-3. 

The terminology becomes incredibly complicated with all the mutants KO mice etc. Given the very nuanced nature of the data the major take home points become hard for the uninitiated to tease out. Potentially a cartoon with each figure may help when interpreting the data so it is easy to visualise where each player sits in this highly complex pathway?

Round 2

Reviewer 1 Report

I thank the authors for their careful implementation of my suggestions. They also answered all questions raised in the original submission.